# A Review of the Utilization of Canola Protein as an Emulsifier in the Development of Food Emulsions

**DOI:** 10.3390/molecules28248086

**Published:** 2023-12-14

**Authors:** Yan Ran Tang, Supratim Ghosh

**Affiliations:** Department of Food and Bioproduct Sciences, University of Saskatchewan, 51 Campus Drive, Saskatoon, SK S7N 5A8, Canada; yanran.tang@usask.ca

**Keywords:** canola protein, rapeseed protein, cruciferin, napin, emulsifier, emulsion formation, emulsion stability

## Abstract

Canola is the second-largest cultivated oilseed crop in the world and produces meal consisting of about 35–40% proteins. Despite this, less than 1% of the global plant-based protein market is taken up by canola protein. The reason behind such underutilization of canola protein and its rapeseed counterpart could be the harsh conditions of the industrial oil extraction process, the dark colour of the meal, the presence of various antinutrients, the variability in the protein composition based on the source, and the different properties of the two major protein components. Although academic research has shown immense potential for the use of canola protein and its rapeseed counterpart in emulsion development and stabilization, there is still a vast knowledge gap in efficiently utilizing canola proteins as an effective emulsifier in the development of various emulsion-based foods and beverages. In this context, this review paper summarizes the last 15 years of research on canola and rapeseed proteins as food emulsifiers. It discusses the protein extraction methods, modifications made to improve emulsification, emulsion composition, preparation protocols, and emulsion stability results. The need for further improvement in the scope of the research and reducing the knowledge gap is also highlighted, which could be useful for the food industry to rationally select canola proteins and optimize the processing parameters to obtain products with desirable attributes.

## 1. Introduction

Canola was developed as a rapeseed cultivar in the 1970s in Canada, with low levels of erucic acid and glucosinolates [1]. Canola refers to the cultivar of the genus *Brassica* (*Brassica napus*, *Brassica rapa* or *Brassica juncea*) from which the oil and the meal should contain less than 2% erucic acid (22:1) and less than 30 μmol glucosinolate per gram of air-dried oil-free solid, respectively [1,2]. Since the high intake of both erucic acid and glucosinolates has been related to negative health impacts, reducing these components improved the usage of canola for edible oil and animal feed purposes, respectively. Canola protein is recognized for its balanced amino acid profile and for providing all essential amino acids. It is particularly rich in the sulphur-containing amino acids cysteine and methionine, whose concentrations reach the reference protein pattern established by the FAO [3]. Various sources of canola proteins showed PDCAAS values as high as 0.81. Moreover, canola protein hydrolysates have been shown to improve the PDCAAS value to 1.00, which shows the potential for canola protein’s use as a valuable nutritional source [4]. The selected hydrolysis of canola protein has also shown its impressive potential as a source of valuable bioactive peptides for human health [5]. Numerous studies have also shown the wide range of techno-functional properties of canola proteins, including emulsification, foaming, gelation, and film formation [3,5], indicating its importance in food and related applications.

However, unlike other plant-based proteins, canola proteins face several unique challenges. First, the high-temperature desolventization step of the commercial oil extraction process significantly damages protein quality. A preferable starting material for improved protein functionality could be cold-pressed meal; however, the presence of a higher amount of residual oil means the need for a cold solvent extraction process. Second, the presence of phenolics in the canola seed and their oxidation during the extraction process negatively affect the colour and flavour of the proteins. Third, the phytates in the canola seed may also bind to the protein, thereby reducing their surface activity [6]. Fourth, two major proteins of canola, cruciferin (a multi-unit globulin) and napin (an albumin protein), have very different amino acid composition, molecular structure, size, and physicochemical properties; hence, the functional properties of canola proteins greatly vary depending on their ratio [7]. These challenges led to the relatively slow utilization of canola proteins as a valuable food ingredient. For example, only a mere 0.14% of the global plant-based protein market is taken up by canola protein [8]. However, canola is the second-largest cultivated oilseed crop in the world after soybean [9,10] and produces a protein-rich meal that would add tremendous value to the crop.

Many food products, such as beverages, coffee creamer, salad dressings, spreadables, and dips, are based on oil-in-water (O/W) emulsions. A food emulsion is formed and stabilized using food-grade emulsifiers. The search for an effective natural emulsifier to replace synthetic emulsifiers, animal proteins, modified starch, and gum Arabic is a long-standing issue in the food and beverage industry. With ever-increasing consumer demand for using all-natural, sustainable, plant-based food ingredients, one of the major challenges in the food industry is to replace all synthetic and animal-origin ingredients with plant-based products. At present, pulse proteins are at the forefront of plant-based emulsifiers due to their lower cost, relatively simple extraction process, wide availability of a variety of pulse sources, popularity and highly efficient emulsification behaviour [11]. This is where canola proteins can play a major role. Although some research has been carried out on the emulsification ability of canola proteins, our knowledge of its proper application in food is still in its infancy, and there is still a vast knowledge gap. In the last 10 years, only about 100 research papers have been published on canola and rapeseed protein-based emulsions. However, to produce high-quality and consistent food products from a chosen ingredient, it is vital to learn how the ingredient compositions and processing affect the product’s physicochemical properties, leading to improved consumer acceptability. In this context, this review paper aims to summarize the last 15 years of research on canola and rapeseed proteins as food emulsifiers, which would benefit researchers. It starts with the composition of canola proteins and their extraction methods, and then the characterization of canola proteins as food emulsifiers. The final section focuses on the physicochemical properties of canola-protein-stabilized emulsions. The need for further improvements in the scope of the research and reducing the knowledge gap is highlighted, which could be useful for the food industry to rationally select canola proteins and optimize the operation parameters to obtain products with desirable attributes.

## 2. Canola Protein Compositions and Minor Components

The major components of canola seed are made up of approximately 40% oil, 20% protein, and 33% carbohydrates, with the rest being moisture and ash [3,12]. After defatting, the protein content in canola meal reaches about 35–40%, where the major constituents are cruciferin (~60%) and napin (~20%) proteins [7,13]. Cruciferin is a 12S hexameric globulin with a molecular weight of around 300–360 kDa, where the six subunits are linked by 12 disulphide bonds. Each subunit comprises one α-polypeptide chain and one β-polypeptide chain [14]. Napin is a 2S albumin with a molecular weight of around 17 kDa, composed of one ~4 kDa and another ~9 kDa polypeptide chain [15]. The polypeptide chains are linked by two inter-chain disulphide bonds, while two intra-chain disulphide bonds are present in the large polypeptide chain [16]. Napin has about 45% of its hydrophobic amino acids mainly located in one distinct domain, whereas in cruciferin, the hydrophobic amino acids are widely distributed across the protein surface [14]. Oleosin is a minor protein (1–4% by weight) present in canola seed, which functions as a stabilizer at the surface of the oil bodies so that the oil remains in the form of discrete droplets in the oil seed [17].

Canola protein is recognized for its balanced amino acid profile and for providing all essential amino acids [3]. A wide range of protein-digestibility-corrected amino acid scores (PDCASS) for canola proteins has been reported (0.61–0.86), which could be due to the source of canola, the extraction method used, and also the assessment model used for the PDCAAS determination [3]. Canola meals are also known to contain certain antinutritional minor components, namely, erucic acid, phenolics, phytic acid, glucosinolates, and some protease inhibitors. Various methods, such as heat treatment [18], ultrafiltration [19], nanofiltration [20], alkali extraction plus membrane processing [21], and protein micellar mass processing [22] have been used to reduce the antinutritional compounds. A recent study on accelerated solvent extraction with high pressure and temperature has enhanced the extraction efficiency of the phenolic antioxidants from canola meal, which could be used for nutraceutical purposes [23]. 

## 3. Canola Protein Extraction Methods

One of the earliest studies regarding rapeseed protein extraction was reported by Porkony and co-workers in 1963, in which the authors precipitated rapeseed proteins from alkaline solutions with diluted acids [24]. Over several decades, different methods have been developed to isolate rapeseed and canola proteins and the cruciferin and napin fractions, for instance, the protein micellar mass process [22,25], calcium precipitation [26], acid and alkali extraction followed by isoelectric point precipitation [27], and electro-activated solution extraction [28]. Though not all studies reported protein yield, Teh, Bekhit, Carne and Birch [27] reported a higher protein yield using the alkali (14.67 ± 0.16%) than the acid extraction (8.00 ± 0.05%) process. Besides precipitation, ultrafiltration has also been employed to extract canola protein. Tzeng, Diosady and Rubin [21] designed a process consisting of the extraction of an oil-free meal at pH 10–12.5, isoelectric precipitation to recover the proteins, and ultrafiltration of the supernatant followed by diafiltration to concentrate and purify the remaining acid-soluble proteins. Through this method, they achieved a protein yield of 42.8% of the starting meal for isoelectric-precipitated proteins and 32.6% for acid-soluble proteins. Fetzer, et al. [29] compared ultrafiltration versus a combination of acidic precipitation followed by ultrafiltration. A higher protein yield (40.4 ± 1.8%) was found using the ultrafiltration method, where the protein was extracted at native pH from the cold-pressed meal. 

The extraction methods mentioned above involve multiple steps, where the canola meal must be defatted first and subsequently subjected to protein extraction. Recently, Ntone and coworkers [30,31] developed an aqueous extraction process for the simultaneous separation of oleosome and protein bodies from rapeseed oilseeds. In this process, de-hulled rapeseeds were dispersed and blended in alkaline water, followed by filtration and centrifugation to obtain oleosomes in the cream layer and the protein bodies and the fibre-rich phase in the serum and the sedimented layers, respectively. The serum layer containing the rapeseed protein was filtrated to obtain a highly soluble protein concentrate. The protein yield in the concentrate was about 31% of the protein in the initial seed, which was claimed to be higher than the conventional isoelectric precipitation from defatted meals (15–28% protein yield) [30].

The two major protein fractions of canola (napin and cruciferin) possess different physicochemical properties and functionalities; hence, researchers tried to separate them from each other to better understand their functional properties, including emulsification behaviour. Since napin and cruciferin have distinct isoelectric points, an integral method can be developed to isolate the two fractions. Wanasundara and McIntosh [32] reported first adjusting the pH to solubilize the napin fraction and obtaining a cruciferin residue. Then, an aqueous extraction was performed on the residue to obtain a soluble cruciferin-rich protein extract and a low-protein residue. An integrated method was also reported by Akbari and Wu [33] to isolate napin and cruciferin fractions from defatted canola meal. This method involved washing the meal at pH 4 to obtain a soluble napin extract and a precipitate, which was adjusted to pH 12.5 to yield an alkaline extract, followed by acidic precipitation at pH 4 to obtain the cruciferin fraction. The supernatant was combined with the napin extract obtained in the earlier step. Ntone, Van Wesel, Sagis, Meinders, Bitter and Nikiforidis [31] used a two-step diafiltration process to isolate the napin fraction from the rapeseed proteins. In the first step, 100 kDa cut-off diafiltration was used to separate the high-molecular-weight non-protein and cruciferin fractions, followed by ultrafiltration and a second diafiltration to remove low-molecular-weight impurities (5 kDa cut-off) to isolate the napin fraction. Studying the emulsification behaviour of napin and cruciferin separately and in combination is important for the improved utilization of canola proteins in emulsion development. However, commercially, such protein enrichment will increase the cost of the ingredient, and unless a significant advantage can be demonstrated, it is better to utilize the whole protein for emulsion development.

## 4. Canola Protein Characterization for Its Emulsification Behaviour

Canola protein has been extensively studied for its various functionalities, such as gelling, emulsifying, foaming, and film-forming properties. In food applications, canola protein can be utilized as an emulsifier in emulsion-based products such as mayonnaise, salad dressing, creamer, beverages, and meat products. In general, several mechanisms are involved during the emulsification process of plant proteins, such as the diffusion of the protein to the oil and water interface, the surface denaturation of the protein to align its hydrophilic moieties to the aqueous phase and the hydrophobic moieties to the oil phase, and, lastly, the steric stabilization provided by the protein to prevent oil droplet destabilization. In this case, the solubility, surface hydrophobicity, and interfacial tension of canola protein are often studied along with their emulsifying properties. It is commonly observed that as the pH increases or decreases away from the isoelectric point, the protein solubility improves [28,34]. Chang, et al. [35] reported a good correlation (r = 0.71; *p* < 0.001) between protein solubility and charge, indicating that the more highly charged proteins are more soluble. Various treatments such as enzymatic hydrolysis [36], ultrasound treatment [37], protein conjugation with dextran [38], and the application of a pulse electric field [39] have also been employed to increase the canola protein solubility, which could favourably impact their emulsification behaviour.

### 4.1. Surface Hydrophobicity

Surface hydrophobicity indicates the amount of non-polar aromatic and aliphatic amino acids on the protein surface [40]. It has more influence on the interfacial properties of globular proteins than total hydrophobicity. Several authors reported an increase in the surface hydrophobicity of canola proteins as the pH decreased from 7 to 3, which was attributed to the protein denaturation at acidic pH, thus exposing the hydrophobic moieties [41,42]. A positive correlation of surface hydrophobicity and emulsion activity index (EAI) (r = 0.642; *p* < 0.01) was also reported [41], which was ascribed to a greater alignment and integration of the protein at the oil–water interface due to the presence of more hydrophobic amino acids on the protein surface [40]. Similar to protein solubility, the surface hydrophobicity of canola protein can also vary depending on the extraction methods and various protein modification treatments, such as hydrolysis [36], ultrasound treatment [37], and conjugation [38], due to the change in the protein conformation and particle size. Intrinsic fluorescence was employed to measure tyrosine and tryptophan fluorescence, which was used indirectly to better understand the protein structure [43]. Tan, et al. [44] reported a negligible change in maximum fluorescence intensity (F_max_) as the pH of the canola protein decreased towards the isoelectric point, while it increased as the pH increased, indicating a more open structure that allowed the hydrophobic groups to be exposed, which favoured emulsion formation [40]. Often, contradicting results of surface hydrophobicity on emulsification behaviour can be found in the literature. Too-high surface hydrophobicity would make a protein aggregate and become insoluble in the aqueous phase, leading to inferior emulsification. In contrast, too-low surface hydrophobicity would make a protein incapable of adsorbing on the oil droplet surface. Hence, an optimum surface hydrophobicity is preferred.

### 4.2. Surface Activity and Interfacial Tension

The interfacial tension of plant protein is studied to examine its effectiveness as an emulsifier. A lower interfacial tension indicates higher surface activity and lower energy required to form stable emulsions. Due to the macrostructure of canola proteins, it takes time for their diffusion and adsorption to the interface; therefore, it is important that interfacial tension values should be taken when it reaches equilibrium, indicated by no further change in interfacial tension at a constant temperature and concentration. Tang et al. (2021) studied the interfacial tension of various concentrations of salt-extracted canola protein isolate (CPI) at pH 7 and found that the time to reach interfacial equilibrium decreased from 15 min for 1 wt% protein to 5 min for 4 wt% protein; however, the equilibrium interfacial tension value was not significantly different (~1.3 mN/m). The literature-reported interfacial tension value of canola protein solution against oil at pH 7 showed great variability among studies, possibly due to various extraction methods, measurement techniques, the presence of oil impurities (such as monoglycerides), and protein concentration. The reported oil–water interfacial tension in the presence of canola protein was 5.24 mN/m (0.2 wt% protein, du Noüy ring) [28], ~14 mN/m (2 wt% protein, du Noüy ring) [35], and 14.8 mN/m (0.25 wt% protein, du Noüy ring) [41]. The air–water interfacial tension was reported as 43.1 mN/m (0.25 wt% protein, du Noüy ring) [45] and ~40 mN/m (0.7 wt% protein, du Noüy ring) [46]. The ratio of the hydrophobic and hydrophilic moieties of the protein could affect its surface-active properties. Therefore, a good balance of both surface hydrophobicity and the solubility of the protein helps decrease the interfacial tension effectively. Though the surface activity of canola protein was often studied using interfacial tension measurement, its behaviour at the interface was not well explored. Krause and Schwenke [6] studied interfacial behaviour with tensiometry and Langmuir–Blodgett techniques. They observed that the interfacial properties of napin dominated the monolayer and emulsion characteristics, while the cruciferin possessed lower surface activity due to its larger size and globular conformation at the surfaces. Ntone, Van Wesel, Sagis, Meinders, Bitter and Nikiforidis [31] also concluded that the adsorption of rapeseed proteins at the oil/water interface was mostly formed by napin due to their smaller size (radius = 1.7 nm), leading to faster diffusion towards the interface and lower energy barrier for surface adsorption due to its unique Janus-like structure, with two distinct domains of hydrophilic and hydrophobic amino acids on the protein surface. Cruciferin, due to its larger size (radius = 4.4 nm), broad distribution of hydrophobic amino acids, and lower surface activity, could not adsorb or displace napin from the surface. Evidence of improved interfacial elasticity in the presence of the mixture of both led to the hypothesis that the weak binding of cruciferin as a secondary layer to the napin primary layer was important for the emulsion stability [31]. However, this hypothesis is in contrast to many other earlier reports that napin could be detrimental to emulsion formation and stability (Akbari and Wu [33],Tan, Mailer, Blanchard and Agboola [44,47]. For example, Wu and Muir [47] found that O/W emulsions (1:5 oil to aqueous phase) prepared with 1 wt% cruciferin protein using a high-speed blender were significantly more stable with a smaller average oil droplet size (1.4 µm) than similar emulsions prepared with napin protein (26.5 µm). The authors proposed that the poor emulsification formation and stabilization properties of napin could be ascribed to its high content of basic amino acids, leading to a lower hydrophobic attraction towards the oil phase. Ntone, Van Wesel, Sagis, Meinders, Bitter and Nikiforidis [31] proposed that napin could adsorb at the interface using its smaller hydrophobic domain; however, for better interfacial stabilization, cruciferin must interact with the adsorbed napin.

### 4.3. Emulsion Stabilization Mechanisms of Canola Protein

The surface activity of canola protein is important for its role in emulsion formation due to its ability to adsorb on the bare droplet surface during homogenization. However, after emulsion formation, the long-term stability of the emulsions could come from the electrostatic repulsion and steric stabilization ability of the canola protein. Electrostatic repulsion relies on droplet charge, which depends on the canola protein extraction method, composition, concentration, and the surrounding environment, such as pH, ionic strength, and temperature. Evidence of electrostatic repulsion can be obtained from the droplet charge. At pH 7, the droplet charge of the canola-protein-stabilized emulsion was reported to range from −15 mV [35] to −26.5 mV [48]. Tang and Ghosh [49] reported a zeta potential of around −12 mV at pH 7 for the O/W emulsion stabilized with salt-extracted CPI, which did not differ significantly as the CPI concentration increased from 1 to 4 wt%. The authors proposed that such low values of zeta potential could be the reason behind the lower electrostatic repulsion leading to droplet aggregation. Interestingly, when the authors added 10% vinegar to the emulsions, the pH was lowered to pH 3.7, and the numerical value of the zeta potential increased to +22 mV, which provided stronger electrostatic repulsion and improved droplet stability against aggregation [49]. Apart from electrostatic repulsion, canola protein’s emulsion stability can also be enhanced by its steric stabilization ability. Steric repulsion occurs due to the intermingling or compression of the interfacial polymeric layer when the two droplets are in proximity. Evidence of the steric stabilization ability of CPI at pH 7 was shown by Tang and Ghosh [49]. The authors added 1 wt% NaCl to the pH 7 CPI-stabilized emulsion, which led to a significant drop in surface charge, but the strong interfacial steric barrier prevented droplet aggregation. Ntone, Van Wesel, Sagis, Meinders, Bitter and Nikiforidis [31] proposed that the primary interfacial layer of napin could be covered with a secondary layer of larger cruciferin at pH 7, which could also be the reason behind the strong steric stabilization ability of canola and rapeseed proteins.

## 5. Physicochemical Properties of Canola-Protein-Stabilized Emulsions

In Table 1 and Table 2, a detailed summary of nearly 20 years of selected research on rapeseed- and canola-protein-stabilized emulsions, respectively, are provided. The tables show details of the protein extraction method, modification, emulsion composition, preparation protocols, emulsion stability results, conclusions, and recommendations. Most research on rapeseed-protein-based emulsions was prepared using low-intensity methods, leading to a coarse, unstable emulsion, which does not help make a sound conclusion (Table 1). Nevertheless, some improvement in the emulsion stability was observed when rapeseed protein was conjugated with dextran [38] or gum arabic [48]. The controlled hydrolysis [50] and acylation of rapeseed peptides [51] have also shown some interesting results on emulsion stability. Several studies on canola proteins showed that cruciferin was a far better emulsifier than napin and whole canola proteins (Table 2). Napin itself appeared detrimental to emulsion formation and stability due to the presence of an excess of basic amino acids, leading to lower amphilicity [33,47]. Some researchers used various protein modification technologies, such as enzymatic hydrolysis [36,52], pulsed electric field assisted extraction [39], and ultrasonication [37], which showed some improvement in emulsion stability. In most research, canola protein was extracted from cold-pressed meal, which is believed to be more functional than conventional meal due to its high-temperature processing. However, no study so far has directly compared the emulsification behaviour of canola proteins extracted from the two different types of meal sources.

Using the information from Table 1 and Table 2, the following sections discuss some specific aspects of rapeseed- and canola-protein-stabilized emulsions, including visual appearance, droplet size, emulsion stability, the effect of protein modification, the influence of various environmental factors, and the rheological behaviour.

### 5.1. Visual Appearance of Canola-Protein-Stabilized Emulsions

Visual appearance is a crucial aspect responsible for the acceptability of food products. The appearance of canola protein isolate has been reported to be yellowish to dark brown due to the presence of phenolic compounds, which form quinones due to oxidation under alkaline extraction conditions, resulting in a brown colour [57]. There were not many reports about the colour analysis of the canola-protein-stabilized emulsion. Wang, Zhang, Chen, He and Ju [54] showed visual images of rapeseed protein nanogel-stabilized O/W Pickering emulsions, which appeared in a darker colour compared to the commonly seen sodium-caseinate-stabilized emulsions, which could be attributed to the yellow to brownish rapeseed protein. We recently used a dark brown coloured CPI to develop a 50% O/W emulsion, which appeared in an off-white colour that could be well utilized in various emulsion-based foods [49]. More research is needed to understand the mechanism of colour development during the processing, emulsification, and storage of canola proteins. Applying colour-masking agents in canola protein emulsion-based foods could also be another way to mitigate the colour issues of canola and rapeseed proteins.

### 5.2. Droplet Size of Canola-Protein-Stabilized Emulsions

Droplet size distribution is important in an emulsion system as it affects the emulsion stability, rheology, turbidity, and palatability [58]. The emulsion droplet size depends on various intrinsic and extrinsic factors, such as protein size, functionality (e.g., interfacial tension), and conformations as influenced by pH, ionic strength, heat, oil-to-protein ratios, the viscosity of the two phases, the presence of other ingredients, the type of emulsification device used, and the processing conditions. The instruments commonly used to make emulsions are high-pressure homogenizers, high-speed dispersers, ultrasonicators, and microfluidizers. In canola protein emulsion studies, the most frequently used emulsification approaches are high-speed dispersion, high-pressure homogenization, and a combination of high-speed dispersion followed by high-pressure homogenization and ultrasonication. To date, no one has reported canola protein emulsions developed with a microfluidizer. The droplet size of canola protein emulsions developed with a high-speed disperser ranged from ~1 μm to about 20 μm [28,33,56], while the ultrasound-sonicated emulsions exhibited a larger droplet size (~17–70 μm) [46,51]. Canola protein emulsions made with a high-pressure homogenizer showed the lowest droplet size (~0.25 μm to ~2 μm) [36,48,54] and the highest stability of all other devices. The droplet size distribution of canola-protein-stabilized emulsion is commonly exhibited as multimodal distribution due to droplet flocculation and protein aggregation in the continuous phase. The variety of methods used for emulsion formation and droplet size measurement made it difficult to compare the results among different studies of canola-protein-stabilized emulsions. 

### 5.3. Stability of Canola-Protein-Based Emulsions

The emulsifying properties of the canola protein isolate-stabilized emulsions were characterized by emulsifying activity (EA), emulsifying capacity (EC), and emulsion stability (ES) or creaming index. It has been reported that napin and a mixture of napin and cruciferin exhibited a higher EC than cruciferin alone [6,29]. It was shown that the cruciferin maintained its globular conformation at the interface and, hence, exhibited lower surface activity. Another reason could be the lower molecular size (mean molecular area: 23 nm^2^) of napin, which allowed it to diffuse faster to the oil and water interface [6]. However, a smaller particle size might not be sufficient to maintain emulsion stability, as a thick interfacial layer is required to stabilize the emulsion droplets from coalescence. In the study of rapeseed protein nanogel particles as Pickering stabilizers, the authors used a confocal micrograph to show that the nanogels produced a dense interfacial film at the droplet surface, generating a steric barrier against coalescence and flocculation [54]. Evidence of a thick interfacial layer using CPI was also reported by Tang and Ghosh [49], where no sign of any coalescence was observed even with extensive droplet aggregated in a 50% O/W emulsion stabilized using 4% CPI.

### 5.4. Modification of Canola Proteins for Improved Emulsification Behaviour

Various approaches have been studied to improve the emulsification behaviour of canola protein. For example, Pirestani, Nasirpour, Keramat, Desobry and Jasniewski [56] investigated the effect of the glycosylation of gum arabic (GA) with canola protein isolate (CPI) on its emulsifying properties. A smaller droplet size was reported for the GA-CPI conjugate as an emulsifier compared to CPI alone and the mixture of both. The improved emulsifying properties were ascribed to the attachment of GA to CPI, rendering a better amphiphilic balance that favours emulsion formation. Gerzhova, Mondor, Benali and Aider [28] compared the emulsification behaviour of the canola protein extracted using an electro-activated solution to those extracted by a conventional alkaline solution; however, the droplet size of the emulsions at pH 7 was not significantly different for the two approaches. However, at pH 4 and pH 9, the canola protein extracted from the electro-activated solution formed smaller droplets than the conventionally extracted protein. Alashi, Blanchard, Mailer, Agboola, Mawson and Aluko [36] investigated the effect of the enzymatic hydrolysis of canola protein on its emulsification behaviour. The resulting emulsions showed a range of droplet sizes depending on the hydrolysate types, hydrolysis period (1 h and 24 h), pH, and storage temperature. However, the lowest droplet size (~2 μm) was achieved for the native canola proteins, while 1 h alcalase-hydrolysate-stabilized emulsions showed a much larger droplet size (~20 μm). Therefore, careful consideration is needed to justify using enzymatic hydrolysis of canola protein for emulsification. Vioque, Sánchez-Vioque, Clemente, Pedroche and Millán [50] reported a higher emulsifying activity achieved by the hydrolyzed rapeseed protein isolate (3.1% degree of hydrolysis) than the unhydrolyzed rapeseed protein, but both the emulsion activity and stability decreased as the degree of hydrolysis increased. The higher degree of hydrolysis generated smaller peptides, resulting in lower efficiency in reducing the interfacial tension. Therefore, both aspects of emulsifying activity and stability must be taken into consideration in the emulsion study.

### 5.5. Influence of Various Environmental Factors on the Stability of Canola-Protein-Based Emulsions

The stability of canola-protein-stabilized emulsion against various environmental factors such as pH, ionic strength, and heat is important in food applications. Modifications such as glycosylation with gum arabic [56], rapeseed protein nanogel as a Pickering stabilizer [54], and dextran–rapeseed protein conjugate and the effect of ultrasound [38] have proven to increase the emulsion stability against pH and thermal treatment. In the study of the glycosylation of canola protein with gum arabic, the conjugate showed higher stability than the CPI at all pH (4–9) values and temperature ranges (30–90 °C), which was ascribed to the inhibition of the unfolded protein aggregation due to the conjugated gum arabic [59]. Qu, Zhang, Chen, Wang, He and Ma [38] also observed improved emulsion stability when heating at 90–100 °C in the rapeseed protein isolate (RPI)–dextran conjugate emulsions compared to the RPI-stabilized emulsions. The introduction of the dextran chain increased both steric and electrostatic repulsions to the proteins, as well as a more unfolded random coil structure, which contributed to their improved emulsifying properties. Recently, Tang and Ghosh [49] investigated the addition of 1 wt% NaCl and 10% vinegar on the stability and inter-droplet interactions in CPI-stabilized 50% O/W emulsions. While at pH 7, droplet aggregation could be prevented with 1 wt% NaCl, under acidic conditions (with 10 wt% vinegar, pH 3.7), extensive droplet aggregation was observed in the presence of 1 wt% NaCl. The authors proposed that due to an increased surface hydrophobicity and lower steric repulsion at acidic pH, canola-protein-stabilized droplets could not withstand the salt-induced charge screening effect, leading to droplet aggregation. Heating the same pH 7 CPI-stabilized emulsions at 80 °C showed a dramatic increase in interdroplet aggregation due to heat-induced protein denaturation, followed by attraction among the exposed hydrophobic groups and the formation of covalent disulphide bonds [49]. However, even with such strong interdroplet aggregation, no sign of coalescence was observed, which indicated the formation of a strong elastic interfacial layer. Such behaviour could be useful in using canola proteins as an effective food emulsifier.

### 5.6. Rheological Properties of Canola-Protein-Stabilized Emulsions

Rheology is an important characteristic in emulsion-based products, for it is responsible for the appearance, flowability, texture, and work needed for handling and processing. However, in the reports of canola-protein-stabilized emulsions, the rheological studies are very limited. It was reported that the canola-protein-stabilized O/W emulsions exhibited a shear-thinning behaviour where the apparent viscosity decreased as the shear rate increased [48,56]. The shear-thinning behaviour is commonly attributed to the disruption of droplet flocculation. As the shear rate increased, the flocculated droplets were disrupted, leading to a lower viscosity. Li, Wang, Dai, Wang, Chen, Ju, Yuan and He [48] reported a decrease in the apparent viscosity of the gum arabic–RPI-conjugate-stabilized emulsions compared to the control RPI-stabilized emulsion at pH 7; however, an opposite trend was observed at pH 8 and pH 9. Pirestani, Nasirpour, Keramat, Desobry and Jasniewski [56] also reported that the viscosity of the gum arabic–CPI-conjugate-stabilized emulsion was higher than the only-CPI-stabilized emulsion. Aluko and McIntosh [52] studied the effect of limited enzymatic hydrolysis of canola proteins in the development of mayonnaise and observed a decrease in the apparent viscosity as the egg yolk was partially substituted (85:15) with unhydrolyzed and hydrolyzed (degree of hydrolysis: 7 and 14%) canola proteins. They ascribed the decreased viscosity to the larger droplet size in the canola-protein-incorporated emulsions [60]. 

Canola protein itself is well known for its gelling properties, as reported by many studies on canola protein hydrogel [61,62,63,64]. However, not much work on canola-protein-stabilized emulsion gels can be found in the literature. Recently, we showed that a viscous 4 wt% CPI (extracted from a cold-pressed meal)-stabilized 50 wt% O/W emulsion could be transformed into a strong elastic emulsion gel by heat-treating at 80 °C for 30 min [49]. The heated emulsion gels showed a storage modulus in the linear viscoelastic region (G′_LVR_) of about 10,000 Pa, which was ten times higher than the unheated emulsions (~1000 Pa). It was proposed that heating resulted in protein denaturation in the continuous phase and at the oil droplet surface and, subsequently, aggregations through hydrophobic interactions and hydrogen bonding, leading to the formation of a 3D droplet network trapping the continuous phase and gelation in the emulsions. The ability to form such strong emulsion gels could provide many exciting opportunities for the food application of canola protein. For example, Tang and Ghosh [65] converted the above-discussed heat-treated 4 wt% CPI-stabilized 50 wt% O/W emulsions into an oleogel by vacuum drying to remove moisture. The emulsion-templated oleogel was then used as a conventional shortening replacer in cake-baking applications. Although the brown colour of the canola-protein-based oleogel led to a darker-coloured cake, the oleogel-based cake was softer with higher springiness than the shortening cake due to the higher cake-specific volume of the former. The authors attributed such behaviour to the larger air channels in the cake stabilized by canola proteins. 

## 6. Conclusions and Recommendations

The lack of utilization of canola proteins as a food emulsifier could be due to the various challenges associated with it, such as the harsh conditions of the industrial canola oil extraction process, the dark colour of the meal, the presence of various antinutrients, the variability in the protein composition, and the different properties of the two major components of the canola proteins. This has led to an underutilization of canola proteins as an emulsifier for various emulsion-based food applications. However, academic research, although scarce, has shown immense potential for canola protein and its rapeseed counterpart in emulsion development and stabilization. In most research, the canola protein was extracted from cold-pressed meal, believed to be more functional than high-temperature-processed conventional meal. However, for commercial success, the proper utilization of conventional canola meal is recommended. Unfortunately, no study so far has directly compared the emulsification behaviour of canola proteins extracted from the two different types of meal sources. There is also a lack of research on utilizing various novel extraction processes on conventional meal to recover highly functional canola proteins for emulsification. Some researchers showed the benefits of protein modification techniques, such as conjugation with a biopolymer or enzymatic hydrolysis; however, the emulsification technique used was insufficient, or the variety of enzymes used and starting meal made the process difficult to compare and fully explore the potential of such modification. Earlier research also showed that the separation of cruciferin and napin fraction could be beneficial for emulsification; however, such an approach, although important for better understanding, would be difficult to justify commercially. In contrast, recent research showed that the utilization of both fractions could improve the interfacial strength and the long-term stability of canola-protein-based emulsions. In fact, researchers have utilized such high interfacial strength of canola proteins in developing heat-induced emulsion gels by forming 3D droplet networks and emulsion-templated oleogels for various food applications. More advanced research is needed to fully utilize canola proteins from various sources (such as cold-pressed vs. desolventizer-toasted meal) as an efficient emulsifier in food and related soft material applications. The authors hope this review article will help inspire the next generation of research on canola protein for its utilization as a valuable ingredient in various emulsion-based food applications.

## Figures and Tables

**Table 1 molecules-28-08086-t001:** Summary of literature on rapeseed-protein-stabilized emulsions: comparison of protein isolation, modification, emulsion composition, preparation, droplet size, and stability. Conclusions and recommendations from the studies are also provided.

Author	Extraction Method and Protein Modification	Emulsion Composition	Emulsion Formation	Emulsion Droplet Size and Stability	Conclusions and Recommendations
Vioque, Sánchez-Vioque, Clemente, Pedroche and Millán [50]	Isoelectric-precipitated rapeseed protein isolate (RPI) from solvent-defatted meal. Alcalase hydrolysis (DH 3.1 to 7.7%) of RPI.	50% corn oil, 7% protein. pH not mentioned.	High-speed homogenization at 10,000 rpm for 2.5 min.	EAI (~50 to 30%) and ES (~70 to 1%) decreased as DH increased from 3.1% to 7.7%.	RPI hydrolysates with the lowest DH significantly improved emulsification. Hydrolysis could be a way to utilize RPI.
Krause and Schwenke [6]	Diethyl-ether-defatted rapeseed flour, aqueous extraction of rapeseed globulin, albumin, and their mixture (chromatographically purified).	40% decane, 0.2% protein. pH not mentioned,	Sonication for 2 min.	EAI: 220 m^2^/g (RPI), 168 m^2^/g (globulins), 418 m^2^/g (albumins), 368 m^2^/g (mixture of globulins and albumin). Highest EAI and smallest droplets were observed for the albumin emulsions.	Albumin fractions were better than the globulins, mixture, and the RPI. Emulsification via sonication may not be industrially relevant.
Sánchez-Vioque, Bagger, Larré and Guéguen [51]	Cold-pressed rapeseed meal hydrolyzed with alcalase. Peptides (average size of 5.6 amino acids) acylated with acyl chloride with C_10_, C_12_, and C_14_ carbon chains.	37.5% n-hexadecane, 0.1% emulsifier, pH 7.	Ultrasonic disruption at 23 kHz for 15 s.	Acylated peptides packed at the interface similar to a small-molecule surfactant, but formed larger droplets. Degree of coalescence decreased with increase in acylation.	Acylation could be an interesting way to impart better surface activity to rapeseed peptides.
Purkayastha, Borah, Saha, Manhar, Mandal and Mahanta [46]	Defatted cold-pressed rapeseed meal, phenolics removed with solvents. Protein isolate extracted via (NH_4_)_2_SO_4_ precipitation. Maleic anhydride acylated rapeseed protein isolate.	30% soybean oil, 0.7% emulsifier, pH 7.	Sonication in the ultrasonic water bath for 10 min.	EC increased (45–80%) and droplet size decreased with an increase in degree of maleylation. ES reached maximum (85%) at 20% maleylation.	Maleylation showed an interesting approach to improve emulsification. But very large droplets due to weak sonication prevents further utilization of the findings.
Qu, Zhang, Chen, Wang, He and Ma [38]	Ethanol-washed meal, isoelectric-precipitated RPI. Protein conjugated with dextran via traditional wet heating and ultrasonication at pH 6 and pH 3.6.	25% soybean oil, 0.2% emulsifier, pH 4–10.	High-speed homogenization at 24,000 rpm for 1 min.	Dextran-conjugated RPI showed improved EAI at pH 4–10 and ES at pH 4–5 and 9–10 compared to the original RPI. Ultrasonic grafting was more efficient than wet-heating grafting in protein functionality.	Dextran conjugation of RPI could be a novel protein modification to improve its utilization in emulsification.
Kalaydzhiev, et al. [53]	Ethanol-treated industrial rapeseed meal to remove phenol and glucosinolate, and then isoelectric precipitation to recover RPI.	5, 10, 15% sunflower and rapeseed oil, 0.25, 0.5, 1.0% RPI. pH 6.	High-speed homogenization at 1000 rpm for 2 min.	Large droplet size for all emulsions. Emulsion stability improved with 1% protein and 15% oil. Higher stability was observed for sunflower oil compared to rapeseed oil.	The difference in emulsion stability for two different oils is interesting, which needs further investigation.
Li, Wang, Dai, Wang, Chen, Ju, Yuan and He [48]	Rapeseed meal defatted using oil press and Soxhlet extraction, isoelectric precipitation to recover RPI. Complex with gum arabic (0–3%) at pH 7, 8, and 9.	10% rapeseed oil, 3% emulsifier. pH 7, 8, and 9.	High-pressure homogenization at 60 MPa for 3 min.	Complex with GA improved emulsion stability at pH 7. Not much improvement at pH 8. 1% GA-RPI was better for improved emulsion stability (droplet size 0.25–0.5 μm) at pH 9.	Thicker interface for RPI-GA improved overall emulsion stability. However, the effect of pH on the type and extent of complexation was not investigated.
Wang, et al. [54]	Isoelectric precipitation of RPI from industrially defatted rapeseed meal. RPI acylated using butanedioic anhydride. Acylated rapeseed protein nanogel (ARPN) prepared by thermal denaturation.	30% rapeseed oil, 0.75% emulsifier. pH: 3.5–8.5. ionic strength: 0–0.4 M.	High-pressure homogenization at 80 MPa for 1 min.	ARPN-stabilized Pickering emulsions stable at pH range 5.5 to 8.5 and up to 0.2 M. salt. Emulsions with 0.5% ARPN or higher remained stable long-term (up to 30 days).	One of the very few papers that showed novel nanogel particles developed from RPI and its utilization in stable food-grade Pickering emulsions.
Ntone, Van Wesel, Sagis, Meinders, Bitter and Nikiforidis [31]	Simultaneous separation of oleosome and proteins from rapeseed oilseeds by blending in alkaline water (pH 9), followed by centrifugation and recovery of proteins from the serum layer.	10% rapeseed oil, 0.2–1.5% protein (1:1 napin: cruciferin), pH 7.	High-pressure homogenization at 250 bars 5 times.	Droplet size decreased with increase in protein, reached a plateau (1.0–1.5 μm) at 0.7%. No change after 7 days. Droplets were aggregated due to low zeta potential (5 mV). Droplet size similar to an equivalent sodium caseinate emulsion.	The authors showed that napin first adsorbs at the interface, followed by weak interaction with cruciferin. Important work highlighting the role of individual proteins.

RPI: rapeseed protein isolate, DH: degree of hydrolysis, EC: emulsifying capacity, EAI: emulsifying activity index, ES: emulsion stability, ARPN: acylated rapeseed protein nanogel.

**Table 2 molecules-28-08086-t002:** Summary of literature on canola-protein-stabilized emulsions: comparison of protein isolation, modification, emulsion composition, preparation, droplet size and stability. Conclusions and recommendations from the studies are also provided.

Author	Extraction Method and Protein Modification	Emulsion Compositions	Emulsion Formation	Emulsion Droplet Size and Stability	Conclusions and Recommendations
Aluko and McIntosh [52]	Alkaline extraction from commercial canola meal. Protease hydrolysis (DH: 7 and 14%) to obtain canola protein hydrolysates (CPH).	Mayonnaise preparation with 10–50% egg yolk replacement using CPH.	Multistep mixing with a mixer at 200 W for 15 min.	Droplet size (d_32_): 5 μm (100% egg yolk), 10 μm (80:20, egg yolk: 7% hydrolyzed CPH), 7 μm (80:20, egg yolk: 14% hydrolyzed CPH).	Hydrolysis improved egg yolk replacement ability of canola proteins. One of the earlier studies of canola proteins.
Wu and Muir [47]	Salt extraction at pH 8 from hexane-defatted canola meal. Cruciferin and napin fractions separated using gel filtration chromatography.	17% canola oil, 1 wt% emulsifier, pH 7 (0.01 M phosphate buffer).	High-speed homogenization for 60 s (speed not mentioned).	Droplet size (d_32_): 9.0 μm (CPI), 1.4 μm (cruciferin), 26.5 μm (napin). Emulsion stability: 90.0% (CPI), 97.8% (cruciferin), 77.4% (napin).	Cruciferin provided better stability than the whole canola protein. The authors proposed napin could be detrimental to emulsion stability.
Tan, Mailer, Blanchard and Agboola [44]	Alkaline extraction of CPI from cold-pressed canola meal, isolation of cruciferin, napin via Osborne method, water, and salt solubilization.	20% canola oil, 1 wt% emulsifier. pH 4, 7, and 9.	High-pressure homogenization at 125 MPa for 3 cycles.	CPI exhibited the lowest EC (400 mL/g). Cruciferin showed the highest EC (1700 mL/g) and EAI (100 m^2^/g), and lowest droplet size (d_43_) (8 μm) at all pH values.	Cruciferin showed the best emulsification behaviour, comparable to commercial soy protein isolate.
Teh, Bekhit, Carne and Birch [27]	Alkaline extraction or acid extraction followed through isoelectric precipitation cold-pressed canola meal. Alkali and acid extracted CPI (Al-CPI, Ac-CPI).	20 and 50% palm oil, 1 and 2 wt% emulsifier, respectively.	Multiple-step high-speed homogenizer at 2000 rpm for a total of 3 min.	EAI: 50% (Ac-CPI and Al-CPI) ES: 100% (Ac-CPI and Al-CPI) Droplet size: 100 μm (Ac-CPI), 85 μm (Al-CPI) Creaming stability: 30% (Ac-CPI and Al-CPI)	Al-CPI was better than Ac-CPI in terms of emulsification. SDS PAGE showed Al-CPI richer in cruciferin.
Cheung, Wanasundara and Nickerson [41]	Salt extraction of cruciferin-rich canola protein isolate from cold-pressed canola seeds.	50% canola oil, 0.25% emulsifier. pH 3, 5, 7, and ionic strength 0–100 mM.	High-speed homogenization at 7200 rpm for 5 min.	Comparable EAI (20 m^2^/g) at pH 3, 5, and 7 at 0 mM salt. With 100 mM salt, EAI decreased with increasing pH. Highest ESI at pH 3, 5, 7 with 0 mM NaCl (15 min). ESI was not affected by pH.	The EAI values were lower than in other studies, possibly due to the very low protein-to-oil ratio used.
Akbari and Wu [33]	Acidic washing (pH 4), alkaline extraction (pH 12.5) of cruciferin, isoelectric precipitation (pH 4), and ultrafiltration to recover napin from commercial canola meal.	17% canola oil, 1 wt% emulsifier.	High-speed homogenization at 24,000 rpm for 1 min.	Droplet size (d_32_): 1.4 μm (cruciferin), 8.9 μm (napin). For cruciferin, ES: 98.7%, EAI: 32.3 m^2^/g, napin emulsions phase-separated after 5 min.	Napin had deteriorative effect on the emulsifying properties of canola protein.
Chang, Tu, Ghosh and Nickerson [35]	Salt extraction of CPI from cold-pressed canola seeds.	10% canola oil, 2 wt% emulsifier. pH 3, 5, 7.	High-speed homogenization at 7200 rpm for 5 min.	Droplet size (d_32_): 5 μm (pH 3), 8μm (pH 5), 15 μm (pH 7) ES: 85% (pH 3), unstable at pH 5 and 7.	For whole canola protein, pH 3 provided better emulsion stability than pH 5 and 7.
Cheung, et al. [55]	Salt extraction of napin-rich canola protein isolate from cold-pressed canola seeds.	50% canola oil, 0.25% emulsifier. pH 3, 5, 7, and ionic strength 0–100 mM,	High-speed homogenization at 7200 rpm for 5 min.	Highest EAI achieved by napin at pH 3, with 50 mM NaCl (23 m^2^/g), highest ESI achieved by napin at pH 3, 5, 7 with no NaCl (16 min).	Very low amount of protein used to stabilize 50% O/W emulsion. Values might not be relevant.
Gerzhova, Mondor, Benali and Aider [28]	Electro-activated (EA) and alkaline extraction from defatted canola meal. EA protein isolate (EAPI) and concentrate (EAPC), alkaline protein isolate (API) and concentrate (APC).	25% canola oil, 1 wt% emulsifier. pH 4, 7, 9.	High-speed homogenization at 7500 rpm for 1 min, then 14,500 rpm for 1 min.	EAI: EAPC and APC showed the lowest EAI. No significant difference in the creaming stability among all emulsions. Emulsion stabilized using EAPI showed the lowest droplet size at pH 9 (20 μm).	Canola protein concentrate performed better at pI, while the isolate was better at high pH values. Not much advantage of electro-activated extraction was seen.
Pirestani, et al. [56]	Alkaline extraction from defatted canola meal. Canola protein isolate (CPI), CPI-gum arabic (CPI-GA) mixture, CPI-GA conjugate.	40% canola oil, 0.7 wt% emulsifier. pH 7 (0.01 M phosphate buffer).	High-speed homogenization at 20,000 rpm for 1 min.	CPI-GA conjugate exhibited the highest EAI (61 m^2^/g) and ESI (70 min) and smallest droplets (d_43_ 8 μm) compared to CPI-GA mixture and CPI.	CPI-GA conjugate significantly improved emulsification of CPI. However, such conjugation may not be desirable considering the push for GA replacement.
Zhang, Wang, Jiang and Qian [39]	Pulsed electric field (PEF)-assisted alkaline extraction of albumin, globulin, and whole canola protein from isopropanol-defatted canola seeds.	50% soybean oil, 4 wt% emulsifier.	Unknown.	PEF treatment improved the EC and ES for albumin, globulin, and whole protein compared to the untreated ones. Highest EC and ES achieved by PEF-treated albumin.	Novel processing changed the protein structure and improved emulsification. However, economic feasibility could be limited.
Alashi, Blanchard, Mailer, Agboola, Mawson and Aluko [36]	Alkaline extraction from commercial canola meal, isoelectric extraction of CPI. CPH prepared with pepsin, trypsin, alcalase, and chymotrypsin hydrolysis (3.5–7.2 g free amino group/100 g CPI).	20% canola oil, 1% emulsifier. pH 4, 7, and 9.	High-pressure homogenization at 10–15 × 10^5^ kPa.	Unhydrolyzed CPI at pH 9 showed highest emulsion stability of all. At pH 7, trypsin-CPH emulsions showed the most stability, and other enzymes also showed comparable stability to CPI. Emulsion stability improved for CPH at pH 4.	Enzymatic hydrolysis could improve canola protein’s emulsification. However, much more research needed to better understand the various factors involved.
Flores-Jiménez, Ulloa, Silvas, Ramírez, Ulloa, Rosales, Carrillo and Leyva [37]	Isoelectric extraction of CPI from commercial canola meal. Ultrasound (40 kHz) treatment of CPI for 0, 15, and 30 min.	50% canola oil, 6.7% emulsifier. pH 2, 4, 6, 8, 10.	High-speed homogenization at 12,000 rpm for 1 min.	Among all pH levels, ultrasound-treated CPI exhibited highest EA (44–55%) and ES (46–55%) at pH 10.	Ultrasound treatment could improve emulsification, but that depends on time and pH. More research is needed for proper utilization.
Tang and Ghosh [49]	Salt extraction of CPI from cold-pressed canola meal.	50% canola oil, 1–4% CPI, pH 7. 10% vinegar (pH 3.7), 1% salt, or both. Heat treatment at 80 °C.	High-pressure homogenization at 188 MPa for 6 cycles.	Droplet size decreased from 16 to 6 μm as CPI increased from 1 to 4%. Droplet aggregation led to a viscoelastic material. Gel strength decreased with addition of salt or vinegar, but increased with both. Heat treatment increased gel strength tenfold.	One of the first studies showing the ability of CPI, from cold-pressed meal, to create strong emulsion gels, stable under various environmental stresses relevant to food.

DH: degree of hydrolysis, CPH: canola protein hydrolysates, CPI: canola protein isolate, EC: emulsifying capacity, EAI: emulsifying activity index, ES: emulsion stability, Al-CPI: alkali-extracted CPI, Ac-CPI: acid-extracted CPI, EA: electro-activated, EAPC: electro-activated protein concentrate, EAPI: electro-activated protein isolate, API: alkaline protein isolate, APC: alkaline protein concentrate, CPI-GA: canola protein–gum arabic, PEF: pulsed electric field.

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
