# Peer review of "A Review of the Utilization of Canola Protein as an Emulsifier in the Development of Food Emulsions"

_molecules, 2023, doi:10.3390/molecules28248086_

Round 1
Reviewer 1 Report
Comments and Suggestions for Authors
This article has concluded all the researches over the past 15 years for the food emulsions of canola and rapeseed proteins. The authors talked about the canola proteins and their minor components, also included the different extraction methods. As emulsifier, the characterization for the canola proteins also has been described in detail. They also discussed the visual appearance, droplet size, emulsion stability, the effect of protein modification, the influence of various environment factors and the rheological behavior of rapeseed and canola protein-stabilized emulsions. The overall impression is that this is a great conclusion for all the work related emulsions of canola and rapeseed proteins, it explained every aspects in detail, also compared different view of research methods and the contrast experiments results which could benefit other researchers a lot. Only several suggestions here. 1) In introduction, the authors talked a lot about the unique challenges, the temperature during the extraction process, the color and flavor of the proteins and low surface activity and the affection of the ratio of cruciferin and napin. But before that, the advantage of canola protein need to be clarified and emphasized. Unless we know how valuable for the canola protein, it will be more helpful for audiences to understand why the work here are essential. 2) Some comparison need to be described in detail, i.e. in section 4.2, the sentence ‘However, this hypothesis is in direct contrast to many other earlier reports that napin could be detrimental to emulsion formation and stability’, it would be worthy to talk about the contrast points in detail from those research.
Author Response
Author's Reply to the Review Report (Reviewer 1)
Comments and Suggestions for Authors
Reviewer 1: This article has concluded all the researches over the past 15 years for the food emulsions of canola and rapeseed proteins. The authors talked about the canola proteins and their minor components, also included the different extraction methods. As emulsifier, the characterization for the canola proteins also has been described in detail. They also discussed the visual appearance, droplet size, emulsion stability, the effect of protein modification, the influence of various environment factors and the rheological behavior of rapeseed and canola protein-stabilized emulsions. The overall impression is that this is a great conclusion for all the work related emulsions of canola and rapeseed proteins, it explained every aspects in detail, also compared different view of research methods and the contrast experiments results which could benefit other researchers a lot.
Authors’ response: Thank you for your evaluation.
Reviewer 1: Only several suggestions here. 1) In introduction, the authors talked a lot about the unique challenges, the temperature during the extraction process, the color and flavor of the proteins and low surface activity and the affection of the ratio of cruciferin and napin. But before that, the advantage of canola protein need to be clarified and emphasized. Unless we know how valuable for the canola protein, it will be more helpful for audiences to understand why the work here are essential.
Authors’ response: We appreciate this comments by the reviewer. We have now added a brief discussion on the value and importance of canola proteins in the introduction. Please see lines 35-47.
Reviewer 1: 2) Some comparison need to be described in detail, i.e. in section 4.2, the sentence ‘However, this hypothesis is in direct contrast to many other earlier reports that napin could be detrimental to emulsion formation and stability’, it would be worthy to talk about the contrast points in detail from those research.
Authors’ response: We have now discussed this contrasting point with data and explanation. See lines 267-276.
Reviewer 2 Report
Comments and Suggestions for Authors
It's a well-organized review that will help guide future studies in this area.
Some minor corrections are marked in the attached file, pls take a look.

Comments on the Quality of English LanguageMinor corrections are highlighted in the attached pdf.
Author Response
Author's Reply to the Review Report (Reviewer 2)
Comments and Suggestions for Authors
Reviewer 2: It's a well-organized review that will help guide future studies in this area.
Authors’ response: Thank you for your evaluation.
Reviewer 2: Some minor corrections are marked in the attached file, pls take a look.
Authors’ response: We have taken all the comments in the attached document and incorporated them into the manuscript. The changes are highlighted.
- Line 67
- Lines 73-76
- Line 98
- Line 116
- Line 158
- Line 204-205
- Lines 216-217
- Line 252
- Line 300
- Line 317-318
- Lines 448